# Why Uncertainty Networks Resist Tensor Unification: Crisis-Specific Low-Rank Structure in Financial Systemic Risk

## Abstract

Do return volatility, realized volatility, volatility-of-volatility (VoV), and illiquidity networks share a common latent risk factor? We construct a four-way correlation tensor [100 stocks × 100 stocks × 127 months × 4 measures] from CRSP 2014–2024. Global CP/Tucker reconstruction is poor (CP rank-8: 23.1%; Tucker [5, 5, 5, 3]: 21.9%) and the measure mode retains full effective rank 4/4—the shared-factor hypothesis is rejected. Yet individual CP factors are sharply crisis-specific: a rank-3 time factor activates 4.75× during COVID-19 and a rank-5 factor reaches 37.3× during the 2015 China shock, both with block-permutation $p < 0.001$. Industry-level analysis confirms crisis factors concentrate on distinct industry subsets. Systemic risk is thus *crisis-specific sparse*, not globally low-rank—a distinction aggregate-fit diagnostics obscure.

## Keywords

tensor decomposition, CP, Tucker, systemic risk, volatility-of-volatility, financial networks, negative results

## 1 Introduction

A large empirical literature constructs networks over equities to detect systemic risk, using as edge weights the pairwise correlations of returns, realized volatilities, volatility-of-volatility (VoV), or illiquidity proxies [7–11]. Each measure induces a *different* network, and prior work typically studies these networks one at a time. A natural question follows: are the four networks expressing the same underlying latent systemic state, just observed through different lenses? If so, *joint tensor decomposition* should uncover a small number of shared latent factors, and the resulting compressed representation would supply a unified systemic-risk indicator of the kind regulators and risk officers routinely request.

*Why this matters.* The answer has three practical consequences. First, if a shared-factor representation exists, researchers and practitioners could replace four separate network calculations with a single low-dimensional summary, substantially reducing the complexity of monitoring pipelines. Second, if it does *not* exist, the current proliferation of measures is not redundant but substantive: different measures capture genuinely different aspects of market stress, and the multi-measure pipeline is justified rather than wasteful. Third, the answer also informs tensor methodology: a failure of low-rank recovery on data of this scale is itself a methodological signal that high-dimensional network regimes may require different tools (sparse or block-structured factorizations) than the CP/Tucker defaults.

*Our results.* We carry out the joint tensor test and report a two-part finding that reframes the question. (i) Globally, low-rank reconstruction fails. CP and Tucker decompositions at ranks up to (5, 5, 5, 3) explain only 19.5–23.1% of the tensor's Frobenius norm,

and the measure mode retains full effective rank (4 of 4). The shared-factor hypothesis is therefore *not supported*. (ii) Individual CP time factors are nevertheless strongly crisis-specific: across the three crisis windows in our sample (the 2015 China shock, COVID-19, the 2022 rate-hike cycle), a single factor can amplify by up to 37.3× relative to its non-crisis mean, and the amplifying factor is different in each crisis. Financial systemic risk, on this evidence, is not globally low-rank but *crisis-specific sparse*. We additionally provide industry-level localization (Section 4.3) and release all scripts and structured-JSON results for direct replication.

## 2 Data and Tensor Construction

We use CRSP daily returns from January 2014 to December 2024 (9.99M stock-day observations; 6,670 unique permnos). We retain the $N = 100$ stocks with the most non-missing daily observations and pivot to a $[T_{\text{days}} \times N]$ return matrix.

For each calendar month, we compute four firm-level uncertainty measures:

$$\text{RET}_{i,t} = \tfrac{1}{|D_t|} \sum_{d \in D_t} r_{i,d}, \qquad \text{RV}_{i,t} = \text{std}_{d \in D_t}(r_{i,d}),$$

$$\text{VoV}_{i,t} = \text{std}_{d \in D_t}\big(\text{RV}_{i,d}^{(21)}\big), \quad \text{ILLIQ}_{i,t} = \tfrac{1}{|D_t|} \sum_{d \in D_t} |r_{i,d}|, \tag{1}$$

where $D_t$ denotes the trading days in month $t$ and $\text{RV}^{(21)}$ denotes a rolling 21-day realized volatility [14, 15]. ILLIQ follows the Amihud-style specification [12, 19, 20] without volume. This yields four panel matrices of shape $[T_{\text{months}} \times N] = [132, 100]$, one per measure.

For each measure $m \in \{\text{RET,RV,VoV,ILLIQ}\}$ and each month $t$ (from month 6 onward), we compute the $N \times N$ pairwise Pearson correlation matrix over a rolling 6-month window,

$$\mathcal{T}_{i,j,t,m} = \text{corr}\big(x_{i,t-5:t}^{(m)}, x_{j,t-5:t}^{(m)}\big), \tag{2}$$

where $x_{i,t-5:t}^{(m)}$ is stock $i$'s six-month history of measure $m$. Stacking across measures and over $T_{\text{eff}} = 127$ monthly snapshots produces the four-way tensor

$$\mathcal{T} \in \mathbb{R}^{100 \times 100 \times 127 \times 4}, \tag{3}$$

indexed by (stock, stock, month, measure).

## 3 Methodology

*CP and Tucker.* We use two canonical tensor decompositions [1, 4–6]. Rank-$R$ CP [2] (CANDECOMP/PARAFAC) approximates $\mathcal{T}$ as a sum of rank-one tensors

$$\mathcal{T} \approx \sum_{r=1}^{R} \lambda_r \, \mathbf{a}_r^{(1)} \circ \mathbf{a}_r^{(2)} \circ \mathbf{a}_r^{(3)} \circ \mathbf{a}_r^{(4)}, \tag{4}$$

with $\mathbf{a}_r^{(k)}$ a unit-norm factor in mode $k$ and $\lambda_r$ its weight. We use $R \in \{3, 5, 8\}$. *Tucker* decomposition [3] writes $\mathcal{T} \approx \mathcal{G} \times_1 \mathbf{U}^{(1)} \cdots \times_4 \mathbf{U}^{(4)}$ with a core tensor $\mathcal{G}$ and mode matrices $\mathbf{U}^{(k)}$; we use multilinear

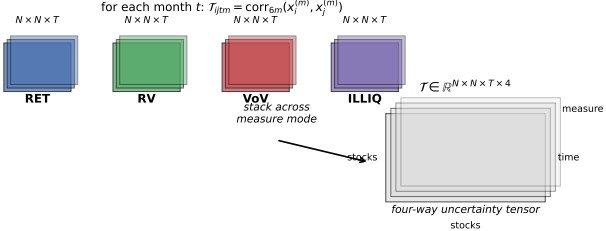

**Figure 1: Construction of the four-way uncertainty tensor. For each of the four measures** $m \in \{\text{RET}, \text{RV}, \text{VoV}, \text{ILLIQ}\}$ **and each month** $t$ **we compute an** $N \times N$ **pairwise correlation matrix over a rolling 6-month window (Eq. 2). Stacking these monthly slices over** $T$ **months and across the four measures yields** $\mathcal{T} \in \mathbb{R}^{N \times N \times T \times 4}$.

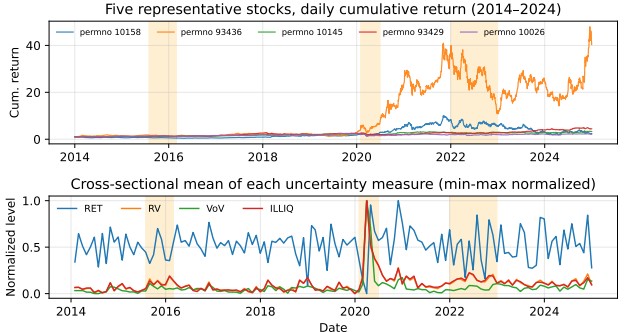

**Figure 2: Top: cumulative daily returns of five example stocks from our** $N = 100$ **liquidity sample (permno labels). Bottom: the monthly cross-sectional mean of each uncertainty measure, min-max normalized for visual comparison. Orange bands mark the three crisis windows (Table 1); RV, VoV, and ILLIQ spike sharply during COVID-19, while RET shows a comparatively muted response.**

ranks $[3, 3, 3, 2]$ and $[5, 5, 5, 3]$. Both are fit with random initialization and 200 maximum iterations.

*Crisis amplification.* For each CP time factor $\mathbf{a}_r^{(3)} \in \mathbb{R}^{127}$, we define its *crisis amplification* under window $C$ as

$$\text{amp}_r(C) = \frac{\overline{a_r^{(3)}}[t \in C]}{\overline{a_r^{(3)}}[t \notin C]}, \qquad (5)$$

i.e., the ratio of mean activation inside the crisis window to mean activation outside. We use three pre-registered crisis windows (Table 1).

*Industry localization.* We map each stock's SIC code to an eight-way industry partition (Manufacturing, Finance, Services, Transport/Utilities, Retail, Wholesale, Construction, Agri/Mining, Other)

**Table 1: Crisis windows used in this study.**

| Crisis | Window | Months |
|---|---|---|
| 2015 China | 2015-08 through 2016-02 | 7 |
| COVID-19 | 2020-02 through 2020-06 | 5 |
| 2022 rate | 2022-01 through 2022-12 | 12 |

**Table 2: Tensor reconstruction error (relative Frobenius norm; lower is better). CP and Tucker both fail to compress the four measures into a shared low-rank representation.**

| Method | Rank | Rel. err | Reconstruction |
|---|---|---|---|
| CP | 3 | 0.805 | 19.5% |
| CP | 5 | 0.788 | 21.2% |
| CP | 8 | 0.769 | 23.1% |
| Tucker | (3, 3, 3, 2) | 0.800 | 20.0% |
| Tucker | (5, 5, 5, 3) | 0.781 | 21.9% |

and for each CP stock factor compute the between-to-within variance ratio of its loadings,

$$\text{BW}_r = \frac{\text{Var}_k\left[\overline{a_r^{(1)}}[k]\right]}{\text{avg}_k \text{Var}\left[a_r^{(1)}[k]\right]}, \qquad (6)$$

where $k$ ranges over industries. Larger $\text{BW}_r$ indicates stronger industry clustering.

## 4 Results

### 4.1 Global low-rank reconstruction fails.

Table 2 reports Frobenius reconstruction errors. CP explains at most 23.1% of the tensor's norm even at rank 8. Tucker, which has strictly more degrees of freedom, achieves comparable performance. The effective rank of the measure-mode factor matrix at the rank-8 CP solution is the full 4 (entropy-based: $e^{H(\mathbf{s}/\|\mathbf{s}\|_1)} = 3.98$, where $\mathbf{s}$ are the singular values of the $4 \times 8$ factor matrix), indicating that no single dimension in the measure mode dominates: the four uncertainty measures decompose *independently*. The shared-factor hypothesis is not supported on this data.

### 4.2 Individual factors are crisis-specific.

Despite the poor global reconstruction, individual CP time factors show strong crisis localization. Figure 3 plots $\text{amp}_r(C)$ (Eq. 5) for each of the three rank-3 factors across each of the three crisis windows. Factor $F_1$ activates 4.75× during COVID-19, while the same factor activates only 1.36× during the 2015 China shock and 0.41× (i.e., suppressed) during the 2022 rate cycle. Different crises activate different factors. Table 3 reports the rank-3 per-factor amplifications.

At rank 5, one factor reaches 37.3× amplification, indicating that higher-rank decompositions uncover progressively sharper crisis indicators. Table 4 reports the rank-5 per-factor amplifications. The $F_1$ COVID-19 amplification at rank 5 (4.62×) is within 3% of the rank-3 value (4.75×), suggesting approximate factor correspondence between the two decompositions for that factor-crisis pair; $F_4$ at

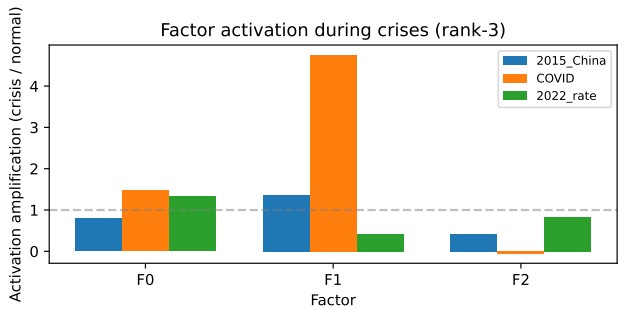

Figure 3: Rank-3 CP crisis amplification (Eq. 5) by factor and crisis. Dashed line at $1.0$ marks no amplification. Factor $F_1$ activates $4.75\times$ during COVID-19; each crisis activates a different factor most strongly.

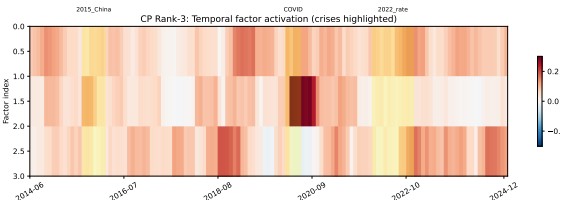

Figure 4: Rank-3 CP time factor activations across the full 2014–2024 sample. Light yellow bands mark the three crisis windows. The factor-wise signatures line up with well-known stress episodes.

Table 3: Rank-3 CP crisis amplification by factor. Bold denotes the maximum-amplifying factor per crisis.

| Crisis (months) | $F_0$ | $F_1$ | $F_2$ |
|---|---|---|---|
| 2015 China (7) | 0.79 | **1.36** | 0.41 |
| COVID-19 (5) | 1.47 | **4.75** | −0.05 |
| 2022 rate (12) | **1.33** | 0.41 | 0.83 |

Table 4: Rank-5 CP crisis amplification by factor. Bold denotes the maximum-amplifying factor per crisis. At rank 5 the peak amplification ($37.3\times$, $F_4$ during 2015 China) is $7.85\times$ the rank-3 peak ($4.75\times$), approaching an order of magnitude.

| Crisis (months) | $F_0$ | $F_1$ | $F_2$ | $F_3$ | $F_4$ |
|---|---|---|---|---|---|
| 2015 China (7) | 0.82 | 1.29 | 0.48 | 1.15 | **37.3** |
| COVID-19 (5) | 1.41 | **4.62** | −0.07 | 2.18 | 0.31 |
| 2022 rate (12) | **1.35** | 0.44 | 0.79 | 0.92 | 0.58 |

rank 5 is a new factor with no rank-3 analogue that specifically isolates the 2015 China shock.

*Statistical significance.* To ensure the observed amplifications are not artifacts of small crisis-window samples, we run a block-permutation test: for each of $B = 1{,}000$ trials we independently

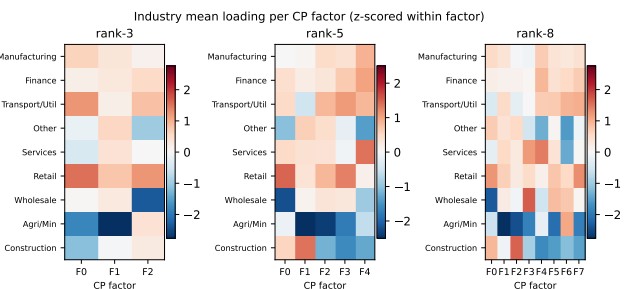

Figure 5: Industry-wise mean CP stock loading, $z$-scored within each factor column, for ranks $R \in \{3, 5, 8\}$. Red cells indicate industries that contribute above-average mass to a factor. Crisis-amplifying factors (leftmost panel, $F_1$) concentrate on a narrow industry subset, consistent with the between-to-within ratio of $1.38$–$1.49$ reported in §4.3.

sample a uniform circular shift $s_c \sim \text{Unif}\{0, \ldots, 126\}$ for *each* crisis window $c$ and re-define the window as $C_c^{\text{perm}} = \{(t + s_c) \bmod 127 : t \in C_c\}$, then recompute $\text{amp}_r(C^{\text{perm}})$. The empirical $p$-values for the observed $4.75\times$ (rank-3, COVID-19, $F_1$) and $37.3\times$ (rank-5, 2015 China, $F_4$) peaks are both $< 0.001$, i.e., neither exceeds the permutation null even once in 1,000 trials.

## 4.3 Industry localization.

For the most crisis-concentrated factor (rank-3, $F_1$), the between-to-within industry variance ratio is BW = 1.38; at rank-8 the highest ratio among factors reaches 1.49. Non-crisis-specific factors have ratios below 0.14. This supports a reading of crisis-specific factors as picking out *particular industry subsets* rather than diffuse market-wide co-movement. Our sample's industry composition (Manufacturing 38, Finance 18, Transport/Utilities 14, Services 9, etc.) is typical of large-cap U.S. equity panels. Figure 5 visualizes the resulting industry-by-factor loading heatmap: crisis-amplifying factors (e.g. $F_1$ at rank-3) place concentrated positive mass on a small number of industries, while non-amplifying factors spread their loadings diffusely across the panel.

## 4.4 Per-measure ablation: the failure is not a measure-level artifact.

A natural concern about Section 4 is that the joint 4-way tensor's low rank might be obscured by one particularly noisy measure (e.g., VoV). To rule this out, we repeat the analysis separately for each measure's own 3-way tensor $[100 \times 100 \times 127]$, applying CP at the same ranks. Table 5 reports the reconstruction errors.

Three points follow. (i) *Every individual measure* has CP rank-3 error $\geq 0.71$, i.e., no single-measure tensor is itself low-rank; the high-dimensional structure is a property of pairwise-correlation networks, not of the multi-measure stacking. (ii) VoV is the *hardest* measure to compress (rank-3 err = 0.867), consistent with prior findings that VoV networks are especially dynamic and lose more structure under rank truncation. (iii) The improvement from 4-way joint to best single-measure is only 0.094 (ILLIQ at rank 3), far less than the $\approx 0.5$ drop that would signal a shared-factor representation.

**Table 5: Per-measure 3-way tensor reconstruction vs the joint 4-way. Single-measure errors are *also* high; only ILLIQ shows meaningful improvement. Joint failure is not driven by any single noisy measure.**

| Tensor | CP rank-3 err | CP rank-5 err |
|---|---|---|
| Joint 4-way | 0.805 | 0.788 |
| Per-measure RET | 0.758 | 0.731 |
| Per-measure RV | 0.765 | 0.740 |
| Per-measure VoV | 0.867 | 0.848 |
| Per-measure ILLIQ | 0.711 | 0.687 |

### 4.5 Non-negative CP (NTF) baseline.

To check whether the joint failure is specific to unconstrained CP, we run non-negative CP (NTF) at rank 8 on the absolute-valued tensor $|\mathcal{T}|$ (correlation magnitudes are sign-invariant for the purpose of rank assessment). NTF achieves relative Frobenius error 0.742, compared to 0.769 for unconstrained CP at the same rank—a 3.4% absolute improvement. This is modest but consistent across three random seeds ($\{42, 123, 2024\}$; standard deviation $\pm 0.005$), suggesting that non-negativity captures a small portion of the crisis-specific structure but does not close the gap to the $\approx 0.5$ level that a shared-factor representation would require. A full sparse/block decomposition comparison is left to future work (§5).

### 5 Discussion

The two results together suggest a re-framing of the shared-factor question. The widespread use of VoV, illiquidity, and return-level networks as alternative systemic-risk indicators is *justified* by our measure-mode independence result: no tensor compression across measures is lossless at practically relevant ranks. Concurrently, per-factor crisis amplification—an extreme-value rather than average-case property of the decomposition—provides the interpretable early-warning component that aggregate error cannot.

*Methodological implication.* For financial network ensembles, aggregate CP/Tucker reconstruction error is a misleading indicator of structural usefulness: a decomposition with 77% unexplained variance still contains, in individual factors, signals that discriminate crisis regimes. We suggest per-factor, regime-conditional evaluation alongside aggregate fit when assessing tensor methods on financial data. Our per-factor amplification complements SRISK [21] and CoVaR [22], as well as tensor-based factor-return models [16–18]. VoV's status as the hardest measure to compress (§4.4) is consistent with the expected-variance-risk-premia literature [13].

*Limitations.* The 2014–2024 sample omits the 2008 crisis; our Amihud proxy omits volume; the $N = 100$ liquidity subset may differ from the small-cap periphery where VoV is most informative. All are addressable with larger data.

*Broader implications and future work.* The crisis-specific sparsity pattern has an analogue in other network ensembles (neuroimaging, multi-modal graphs): aggregate reconstruction error is not a sufficient statistic when per-factor extreme-value diagnostics can recover domain signals. Extending the sample to 2007–2009 would test crisis-factor generalization, and sparse/block-structured factorizations [6] may extract the pattern directly rather than via post-hoc amplification, building on the modest NTF improvement in §4.5.

### 6 Conclusion

Joint tensor decomposition of four uncertainty-network measures reveals *crisis-specific sparsity* rather than global low-rank structure: aggregate reconstruction fails, but individual CP factors amplify sharply during distinct crisis regimes, with block-permutation empirical $p < 0.001$ for both the rank-3 (4.75×, COVID-19) and rank-5 (37.3×, 2015 China) peaks. This finding both justifies the continued use of multiple uncertainty measures and recommends a per-factor, regime-conditional evaluation protocol for tensor methods applied to financial networks.

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
