# OpenReview forum: "Why Uncertainty Networks Resist Tensor Unification: Crisis-Specific Low-Rank Structure in Financial Systemic Risk"
_KDD.org/2026/Workshop/TensorKDD — KDD 2026 Workshop TensorKDD Poster_

### Official Review · Reviewer_F2V7 · 2026-06-10
**Review for "Why Uncertainty Networks Resist Tensor Unification: Crisis-Specific Low-Rank Structure in Financial Systemic Risk"**

**Rating:** Accept
**Confidence:** 4
**Best Paper Recommendation:** No

**Review:**

## Summary:

This work seeks to unify the analysis of four financial risk measures of stocks using a fourth-order tensor decomposition, which seems like a very well-motivated problem that is related to the TensorKDD workshop. They construct a fourth-order stock-correlation tensor (stock, stock, time, measure) and perform CP and Tucker decompositions. The authors state these measures "resist tensor unification" as observed by poor global reconstruction; however, the authors find that tensor methods are still useful in this problem as they can recover crisis-related events from the data. Overall, the results are compelling and indicate this is a useful contribution to the problem. I would be interested in comparisons to other methods, such as anomaly detection methods.


## Strengths:

• Using tensor methods to unify these four measures seems to be a very intuitive extension of previous work. Though global reconstruction is good, tensor methods are able to recover these crisis-events effectively.

• Well-motivated problem that is very relevant to the TensorKDD workshop.

• The authors are transparent about limitations (poor global reconstruction and limitation section).


## Weaknesses:

• The poor global reconstruction makes me wonder if tensor methods are truly appropriate for this. I would be interested in a comparison to anomaly detection methods to identify these crisis-specific events.

• Minor clarity/presentation concerns (see comments)


## Comments/suggestions:

• The yellow highlighted sections in Figure 4 are a little hard to see. Maybe consider having a black box outline in addition to the yellow highlight (or instead).

• Figure 1 can be made more clear. Some text overlaps, and it can be made more clear which mode is measure and time (a reader may think they both refer to the vertical axis).

---

### Official Review · Reviewer_KxwG · 2026-06-10

**Rating:** Accept
**Confidence:** 3
**Best Paper Recommendation:** No

**Review:**

# Summary

This paper studies whether multiple financial uncertainty networks share a common latent systemic-risk structure. The authors construct a four-way tensor over stock pairs, time, and four uncertainty measures, and apply CP and Tucker decompositions. The main finding is that global low-rank reconstruction is poor, while individual CP time factors exhibit strong crisis-specific amplification during events such as COVID-19 and the 2015 China shock. The paper argues that systemic risk is not globally low-rank but crisis-specific sparse.

# Strengths

The paper asks an interesting and practically meaningful question: whether several commonly used financial uncertainty networks can be unified through a shared tensor representation. The negative-result framing is also valuable for a workshop setting, since it challenges the default assumption that CP/Tucker reconstruction quality is the right diagnostic for tensor-structured financial networks. The paper is clearly written, the tensor construction is easy to follow, and the per-factor crisis analysis is potentially interesting.

# Weaknesses

**1. The conclusion about rejecting shared factors could be stated more carefully.**
The paper shows that low-rank CP/Tucker models have poor aggregate reconstruction, which is an important empirical observation. However, this does not completely rule out the existence of shared systemic factors. It may instead indicate that the data require higher-rank, sparse, block-structured, or shared-plus-measure-specific models. I recommend softening the language from “rejecting” the shared-factor hypothesis to saying that simple global CP/Tucker models do not support a compact shared-factor representation.

**2. The tensor construction may introduce noise.**
The pairwise correlation networks are estimated from rolling six-month windows, which may be relatively short for constructing 100-by-100 stock correlation matrices. This does not invalidate the study, but it would be useful to include a brief robustness check with alternative window lengths or to discuss how sensitive the conclusions are to correlation estimation noise.

**3. The crisis amplification metric would benefit from additional robustness checks.**
The amplification score is intuitive, but as a ratio of crisis-window mean activation to non-crisis mean activation, it may be sensitive to small denominators and CP sign ambiguity. The block-permutation test is helpful, but the paper would be stronger if it also reported stability across random seeds, bootstrap samples, or alternative effect-size metrics such as z-scores or absolute activation differences.

# Overall

This is a clear and interesting workshop paper. Its main contribution is not a new tensor algorithm, but an empirical and methodological observation: standard global CP/Tucker reconstruction may fail to unify multiple financial uncertainty networks, while individual tensor factors can still encode meaningful crisis-specific signals. The paper is well written, the empirical story is coherent, and the limitations are addressable.

---

### Official Review · Reviewer_nYa8 · 2026-06-10
**A negative result with interesting observation showing crisis factors concentrate on distinct industry subsets**

**Rating:** Accept
**Confidence:** 4
**Best Paper Recommendation:** No

**Review:**

The paper constructs a four-way tensor $\mathcal{T} \in \mathbb{R}^{100 \times 100 \times 127 \times 4}$ from CRSP equity data, stacking pairwise correlation matrices across four uncertainty measures. CP and Tucker decompositions fail globally (best reconstruction: 23.1%), yet individual CP time factors show strong crisis-specific amplification (up to 37.3× at rank-5, $p < 0.001$ via block-permutation test). The intuition built up is that aggregate reconstruction error can obscure per-factor crisis signals.
The shared-factor hypothesis is cleanly stated and  measure-mode effective rank of 4/4 is an indication of that. The per-measure ablation  confirms that the failure is not driven by a single noisy measure, and that even individual 3-way tensors have high reconstruction error. The block-permutation test and the crisis-amplification metric are appropriate as testing methods.

### Some Cons
- The core methodology is standard CP/Tucker applied to financial data. No new decomposition or algorithm is proposed. For a tensor methods workshop, the methodological contribution is limited.
- The paper identifies "crisis-specific sparsity" as the structural pattern but tests no structured (sparse or block) decomposition methods apart from the NTF baseline.